# The Landscape of Point Mutations in Human Protein Coding Genes Leading to Pregnancy Loss

**DOI:** 10.3390/ijms242417572

**Published:** 2023-12-17

**Authors:** Evgeniia M. Maksiutenko, Yury A. Barbitoff, Yulia A. Nasykhova, Olga V. Pachuliia, Tatyana E. Lazareva, Olesya N. Bespalova, Andrey S. Glotov

**Affiliations:** Department of Genomic Medicine, D.O. Ott Research Institute of Obstetrics, Gynaecology and Reproductology, Mendeleevskaya Line 3, 199034 St. Petersburg, Russia; jmrose@yandex.ru (E.M.M.); yulnasa@gmail.com (Y.A.N.); for.olga.kosyakova@gmail.com (O.V.P.); lazata1997@gmail.com (T.E.L.); shiggerra@mail.ru (O.N.B.)

**Keywords:** miscarriage, pregnancy loss, spontaneous abortion, RPL, genetic variant, mutation

## Abstract

Pregnancy loss is the most frequent complication of a pregnancy which is devastating for affected families and poses a significant challenge for the health care system. Genetic factors are known to play an important role in the etiology of pregnancy loss; however, despite advances in diagnostics, the causes remain unexplained in more than 30% of cases. In this review, we aggregated the results of the decade-long studies into the genetic risk factors of pregnancy loss (including miscarriage, termination for fetal abnormality, and recurrent pregnancy loss) in euploid pregnancies, focusing on the spectrum of point mutations associated with these conditions. We reviewed the evolution of molecular genetics methods used for the genetic research into causes of pregnancy loss, and collected information about 270 individual genetic variants in 196 unique genes reported as genetic cause of pregnancy loss. Among these, variants in 18 genes have been reported by multiple studies, and two or more variants were reported as causing pregnancy loss for 57 genes. Further analysis of the properties of all known pregnancy loss genes showed that they correspond to broadly expressed, highly evolutionary conserved genes involved in crucial cell differentiation and developmental processes and related signaling pathways. Given the features of known genes, we made an effort to construct a list of candidate genes, variants in which may be expected to contribute to pregnancy loss. We believe that our results may be useful for prediction of pregnancy loss risk in couples, as well as for further investigation and revealing genetic etiology of pregnancy loss.

## 1. Introduction: Miscarriage and Genetics

Human infertility is a frequently occurring problem, affecting more than 50 to 80 million couples worldwide. Despite advances in diagnosis and treatment, the disease etiology remains unexplained in more than 30% of cases, strongly suggesting the involvement of genetic, epigenetic, and environmental factors. Besides inability to conceive, many couples face reproductive problems during pregnancy and childbirth, including pregnancy loss.

According to the European Society of Human Reproduction and Embryology (ESHRE), pregnancy loss (PL) is the spontaneous demise of a pregnancy before the fetus reaches viability [1]. Many genetic studies, however, jointly study cases of unintentional pregnancy loss and pregnancy termination for fetal abnormality (hence, we will use the term ‘pregnancy loss’ to refer to both of these cases). Unintentional pregnancy loss includes spontaneous abortion (SA or miscarriage) defined as fetal death prior to 20 weeks of gestation (though the exact threshold may vary in different sources and countries), and stillbirth defined as fetal death at a later period of gestation [2]. Besides spontaneous pregnancy loss, recurrent pregnancy loss (RPL) is an important reproductive health problem and affects 2–5% of couples. The incidence of RPL may vary between reports because of the differences in the definitions and criteria used, as well as the populations’ characteristics [3]. Back in 1976, the World Health Organization (WHO) defined the RPL as three and more consecutive miscarriages before the 22nd week of gestation, or the loss of a fetus weighing <500 g [4]. Later, in 2011, in line with the WHO definition, the Royal College of Obstetricians and Gynecologists (RCOG) guideline defined recurrent miscarriage as the loss of three or more consecutive pregnancies before 24 weeks of gestation, without imposing any limits on the fetal weight [5].

Miscarriages are common among both parous and nulliparous women, with about 43–49% of women reporting having experienced one or more spontaneous miscarriages. One in every seventeen parous women have three or more miscarriages [6,7]. Multiple factors are involved in miscarriage, such as genetic variation, structural abnormalities and infection of the reproductive tract, endocrine factors, immunological factors, drug, poison, maternal systemic diseases, and so on [8,9]. In addition to sporadic miscarriages, termination of pregnancy for fetal anomaly happens at a rate of 4.6 per 1000 births (according to European statistics). In the UK, over 70% of congenital anomalies are detected during pregnancy, and of those around 37% result in termination [10].

Similar to spontaneous miscarriage, multiple factors contribute to RPL, including maternal age (9–75% of RPL cases), endocrine diseases (17–20%), uterine morphological pathologies (10–15%), chromosomal abnormalities (2–8%), thrombophilia, infectious agents (0.5–5%), and autoimmune disorders (20%). Nevertheless, in approximately 50–75% of RPL cases, the exact cause is not clearly identified and therefore remains unexplained (idiopathic) [5].

Determination of the underlying genetic factors involved in spontaneous or recurrent pregnancy loss currently includes cell culture and karyotype analysis, as well as chromosome microarray analysis (CMA) of chorionic villus tissues. In clinical practice, SNP array analysis is sometimes applied for detecting causal copy number variation (CNV) [11,12]. Previous studies showed that 70–80% of sporadic spontaneous abortions were caused by an abnormal embryonic karyotype, with embryonic aneuploidies being one of the most frequent causes of miscarriage before 10 weeks of gestation [13]. Nevertheless, the underlying cause of the condition is not determined by current standard-of-care practices for a substantial number of cases. In RPL, genetic factors such as DNA methylation, sperm DNA fragmentation, chromosome heteromorphisms, and single nucleotide genetic variation, have been shown to play a role in pathogenesis of the condition. However, none was proven to be a stand-alone risk factor for RPL [5].

In this work, we tried to systematically review recent progress in the investigation of the genetic factors of pregnancy loss, including miscarriage, stillbirth or termination of pregnancy for morphological anomalies. In particular, we analyzed the results of studies that have reported point mutations contributing to loss of euploid pregnancies to understand the common properties of human genes involved in these conditions, as well as to compile a list of candidate genes for future research in this field. Taken together, the knowledge of specific genes that contribute to pregnancy loss could be of importance in designing a diagnostic sequencing panel for affected couples, as well as in understanding the biological pathways that can cause any type of reproductive loss.

## 2. Genetic Research into Pregnancy Loss: The Evolution of Methods

A correlation between chromosomal abnormalities and spontaneous abortion has been observed since the 1960s [14]. In the 1970s, Boue et al. published one of the earliest large cytogenetic studies where almost 1500 samples of fetal tissue were karyotyped, and a chromosomal abnormality rate of over 60% was found [15,16]. Currently, cytogenetic evaluation of the products of conception (POCs) remains one of the most important approaches to determine the genetic cause of pregnancy loss. Cytogenetic studies have revealed that fetal chromosome abnormalities account for at least half of the cases of SA before 12 weeks and nearly a third in the second trimester miscarriages. Most of these abnormalities comprise numerical chromosomal aberrations (86%), whereas a minority of the cases show structural chromosomal abnormalities (6%) and chromosome mosaicism (8%) [17]. The cytogenetic method is helpful to estimate the recurrence risk and give advice for subsequent pregnancies. However, this approach still had many challenges to overcome. Cytogenetic studies of SA or intrauterine fetal deaths rely on conventional tissue culture and karyotyping. This technique has known limitations such as culture failure and selective growth of contaminating maternal cells [18,19].

The technologies for miscarriage analysis have become more and more accurate in the last few decades. Later emerging techniques such as Chromosomal Comparative Genomic Hybridization (CGH), array–Comparative Genomic Hybridization (array-CGH), Fluorescence in situ hybridization (FISH), Multiplex Ligation-dependent Probe Amplification (MLPA), and Quantitative Fluorescent Polymerase Chain reaction (QF-PCR) have overcame some of the disadvantages inherited from conventional cytogenetic techniques, including poor chromosome preparations, culture failure, or maternal cell contamination. These molecular biological techniques offer multiple advantages, including short turnaround time and high resolutions.

CGH uses different fluorescent tags to label control (reference) DNA and test DNA which are then competitively hybridized to metaphase chromosomes. This approach could be used for screening imbalances in the whole genome in a single experiment, and its accuracy and usefulness have been demonstrated in cancer cytogenetics [20,21]. However, CGH is unable to differentiate between diploid, triploid and tetraploid states. Thus, analysis of the ploidy status is sometimes performed by the complementary use of flow cytometry [22] or FISH [23,24,25]. Submicroscopic chromosomal anomalies also may be diagnosed using FISH, but this is highly limited in being a ‘targeted’ approach, requiring knowledge that a specific karyotypic rearrangement is associated with a particular embryonic phenotype [26].

aCGH is a modification of CGH which, instead of using metaphase chromosomes, uses slides arrayed with small segments of DNA as the targets for analysis. This method is used both in PL studies and prenatal diagnosis [27]. A recent systematic review reported an increase in prenatal detection rate of 10% (95% confidence interval (CI), 8–13%) from using aCGH compared to conventional karyotyping in fetuses with structural malformations [28]. In the genetic research of pregnancy loss, microarray-based chromosome analysis has been applied to numerous cases. For example, one group of researchers have provided summary of findings from over 3000 miscarriages that underwent chromosome microarray analysis [29,30,31]. Clinically relevant CNVs were identified in only 1.6% of cases [32], while rare variants of uncertain clinical significance (VUS) were seen in up to 40% of cases [33]. However, despite its common use, aCGH is limited to the detection of CNVs of >10–100 kb in specifically targeted regions [26].

Despite the benefits of the aforementioned genetic testing methods, the etiology of 10 to 60% of spontaneous abortions remains unexplained despite extensive investigation [34,35]. It can be hypothesized that the majority of these cases are caused not by larger chromosomal aberrations or copy-number variants, but by other, smaller-scale genetic events undetectable by these methods. For a long time, there has not been enough evidence supporting the role of such factors as single-gene abnormalities, uniparental disomy, or genomic imprinting in the pathogenesis of miscarriages. Today, however, clinicians are paying more and more attention to the causal single nucleotide substitutions, as well as short insertions and deletions (indels) that may cause pregnancy loss [36]. This change is facilitated by the emergence and rapid spread of Next Generation Sequencing (NGS) technologies that allow for reading the entire sequence of the patient’s genome (or its most relevant parts).

NGS has markedly enhanced reproductive medicine, improving prenatal diagnostic yield by identifying pathogenic genetic variants that are below the resolution of aCGH and other methods in clinical use, as well as by localizing breakpoints of cytogenetically balanced chromosomal rearrangements to individual genes [37]. More recently, NGS has become a necessary instrument for studying the genetic basis of spontaneous abortions and RPL [33], discovering new variants which are potentially related to pathogenesis of these conditions [38]. However, incomplete understanding of the molecular pathology of PL may hinders the identification of the genetic causes of PL by NGS.

In practice, NGS can be combined with previously used chromosome microarray analysis (CMA or aCGH), which opens new possibilities for a comprehensive screening of the genetic causes of pregnancy loss. Sanger sequencing is also commonly applied for validation of potential causal genetic variants, identified by NGS methods [38,39,40].

## 3. Next Generation Sequencing in the Analysis of Pregnancy Loss Genetics

Several NGS-based strategies are routinely used in medical genetics, differing in their costs and diagnostic yield. Whole-genome sequencing (WGS) is the most comprehensive of such strategies. In the field of pediatric genomics, a diagnostic yield of 42% has been reported for trio-based WGS [41]. However, WGS is expensive and demanding in terms of resources needed for data analysis, storage, and interpretation. Hence, alternative targeted strategies are commonly used, such as whole-exome sequencing (WES) and clinical exome sequencing.

Exome sequencing methods are focused on the coding part of the genome called ‘exome’, which makes up ≈1% of the human genome, and is estimated to contain more than 80% of the genetic mutations associated with disease [42]. WES was first introduced in 2009 and adopted quickly as a highly effective approach for postnatal and prenatal genetic diagnosis of Mendelian disorders. WES has been shown to provide a valuable diagnostic option in postnatal genetic evaluation because it is not disease- or gene-specific and does not require prior knowledge regarding the potential causal gene(s) for an observed phenotype [43]. Exome sequencing has therefore started to be incorporated into clinical care for pediatric and adult populations. Generally, WES is the preferred method of sequencing because it is cheaper than WGS and has a smaller, more manageable data set while still comprehensively covering the coding regions of DNA [36]. In terms of prenatal diagnosis, if karyotype testing and CMA cannot determine the underlying cause of fetal malformations and structural abnormalities, WES can provide relevant information to aid in current pregnancy management. Currently, WES on the products of miscarriage is helpful to identify lethal genetic variants in key genes, and it expands our knowledge of prenatal phenotypes of many Mendelian disorders [44]. However, multiple studies have been conducted over the years to demonstrate the utility of NGS methods for this purpose.

In 2013, it was hypothesized that exome sequencing was able to detect underlying genetic factors of pregnancy loss, uncovering the association between miscarriage and single or multigenic changes. Shamseldin et al. published the first report of WES on a family with recurrent pregnancy loss due to nonimmune hydrops fetalis, through which they identified a homozygous rare variant in a highly conserved region of the *CHRNA1* gene as a Mendelian cause [45]. After this proof-of-concept report for a single case, WES-based studies that included several (or even dozens of) fetuses from different families began to appear. In 2014, Carss et al. performed exome sequencing on 30 non-aneuploid fetuses and neonates with diverse structural abnormalities detected by prenatal ultrasound. They identified candidate pathogenic variants for some of the cases, concluding that exome sequencing may substantially increase the detection rate of underlying etiologies of prenatal abnormalities. In 3 out of 30 fetuses, they found highly likely causal variants in *FGFR3* and *COL2A1* genes, as well as a de novo 16.8 kb deletion that included most of *OFD1* [26]. In a further proof-of-principle study, a strategy was developed to diagnose rare autosomal recessive lethal disorders by exome sequencing in non-consanguineous couples with a history of multiple affected fetuses. The aim of the study was to obtain a molecular genetic diagnosis and enable prenatal testing in current/future pregnancies. In this work, heterozygous *DYNC2H1* variants were successfully identified as causing short-rib polydactyly (leading to pregnancy termination). Another two families presenting with a current at-risk pregnancy were then studied prospectively, and a molecular genetic diagnosis was obtained in both families through the identification of *GLE1* and *RYR1* variants causing a severe form of fetal akinesia syndrome with arthrogryposis [46].

Further application of WES detected relevant alterations in four out of seven cases of fetal demises in a cohort of American patients [47] and compound heterozygous mutations in *DYNC2H1* and *ALOX15* in two out of four miscarriages among families with recurrent pregnancy loss [33].

In the latest studies, many genetic variants related to pregnancy loss have been found by WES, further proving the contribution of point mutations to the etiology of pregnancy loss. A 2017 WES-based study detected definitely and likely causal variants in 20% and 45% of the 84 fetal death cases with ultrasound anomalies, respectively. The most frequently reported ultrasound anomalies were central nervous system abnormalities (mutations in *PIK3CA*, *FLNA*, *AMER1*, *PIK3R2*, and *L1CAM* genes) and hydrops/edema (affected genes: *PIEZO1*, *HRAS*, *RIPK4*, *FOXP3*, *MRPS22*, *CYP11A1*, *RIT1*). Abnormalities in the cardiovascular system were also observed frequently, though pathogenic or likely pathogenic variants were detected only in a single case in the *FANCB* gene [48]. In 2018, Mengu Fu et al. performed exome sequencing on 19 products of miscarriage of unrelated couples and reported 36 rare variants in 32 genes associated with miscarriage. Gene Ontology analysis of these genes revealed the enrichment of biological processes involved in early embryonic development, including ‘chordate embryonic development’, ‘cell proliferation’ and ‘forebrain development’ [49].

In another large-scale study, targeted sequencing on a panel of 70 genes associated with cardiac channelophies and cardiomyopathies detected pathogenic variants in 12% of 290 cases of stillbirth in Stockholm County [50]. A greater diagnostic yield was reported by Najafi et al. in an exome sequencing study of Iranian RPL cases (25–40%) in whom oligoarray CGH was normal and the maternal causes of miscarriage were ruled out. In most cases, there were no phenotypes other than lethality, and a possible variant was found in 65% of cases. In 45% of cases, the variant was classified as pathogenic/likely pathogenic according to the American College of Medical Genetics and Genomics (ACMG) guidelines [51]. Despite the differences in case selection criteria and inconsistency in variant classification from these studies, accumulated research data indicated that exome sequencing is instrumental in identifying monogenic causes of a significant portion of pregnancy loss cases and should be integrated into the diagnostic practice [52].

The largest NGS-based study conducted to date was published in 2023 by Byrne et al. [34]. In this work, the ‘genomic autopsy’ using exome and genome sequencing was performed as an adjunct to standard autopsy for miscarriages in 200 families. For 105 of these families who had experienced fetal or newborn death, a definitive or candidate genetic diagnosis was provided, and novel phenotypes and disease genes were described. More importantly, the study showed evidence of severe atypical in utero presentations of known genetic disorders. Pathogenic (P) and likely pathogenic (LP) variants in disease-associated genes were identified in 42 of 200 families (21%). For an additional 10 of 200 families, ACMG-classified VUS were detected, for which proof of pathogenicity was obtained using experimental methods. The majority (57.7%, 30 of 52) of variants leading to definitive diagnoses occurred de novo in the proband. In addition to the 52 families, candidate variants with no additional support of pathogenicity were discovered for further 53 families: (1) novel variants (17 of 53) or phenotypes (10 of 53) not previously described in well-established OMIM disease genes, and (2) predicted deleterious variants (26 of 53) in genes with none to limited evidence of gene–disease relationships.

## 4. A Systematic Review of Pregnancy Loss Genes

As demonstrated in the previous section, the focus of genetic research in pregnancy loss has shifted over the last decade from single cases to large-scale NGS-based studies. Since then, much information about causal genes and genetic variants in these genes has accumulated, but has not yet been systematically aggregated. To address this issue, we analyzed the results obtained in 31 different studies that were found in PubMed using a set of keywords presented in Figure 1. The analysis was restricted to studies which employed NGS methods for causal variant identification.

First, we compiled a list combining all 270 genetic variants found in these works, with these variants scattered across 196 unique genes (the complete list is presented as Appendix A). The dataset included genetic variants observed in euploid fetuses at all gestation ages (the exact gestation age is reported in 57% of cases, with approximately one third of these (36%) corresponding to first-trimester miscarriages). This set of variants included variants of three classes defined according to the ACMG criteria [53]—pathogenic (P), likely pathogenic (LP), and VUS. Different strategies for exome sequencing were used to identify these variants, including fetus-only, trio-, and quad-based analysis.

We next divided the genes and variants into categories according to spontaneity and recurrence of pregnancy loss reported in each case. Most of the variants were associated with miscarriage and stillbirth (147 unique genes/184 variants), and ≈25% of cases corresponded to willful termination of pregnancies due to developmental defects observed on ultrasound (47 genes/63 variants). For 23 variants, ’fetal demise or termination’ was stated as a cause of pregnancy loss, but the exact reason was not stated for each case. Among all cases, 120 distinct variants in 90 genes were reported in recurrent pregnancy loss, while the remaining 150 mutations in 117 genes were causal for a single pregnancy loss, or information about recurrence was not provided in the original study (Figure 1).

During the analysis of these variants, we discovered 57 genes in which mutations were identified two or more times; among them, 18 genes were mentioned in independent studies as causal for pregnancy loss. The genetic variants identified in these genes in various studies are summarized in Table 1. Further analysis of the functions of these genes, as well as the clinical features of the associated pregnancy loss cases, suggested that these genes could be broadly divided into several categories: (i) early embryonic development genes with mutations leading to early lethality; (ii) genes involved in structural development of the embryo, mutations in which cause serious abnormalities that could be detected by ultrasound and can lead to both spontaneous abortion and termination; (iii) genes with important tissue-specific functions, mutations in which could occasionally lead to pregnancy loss and (iv) genes that regulate the structure and function of the uterus or placenta. Below, we briefly summarize the functions of these genes and their role in pregnancy loss.

In total, 2 of the 18 genes exemplify the first group, *PADI6* and *STIL*. *PADI6* variants have no reported postnatal phenotype; nevertheless, homozygous nonsense mutations (e.g., c.1141C>T (p.Gln381*)) were shown to be responsible for early embryonic arrest. Compound heterozygous variants in *PADI6* were also identified in embryos that displayed developmental arrest after an in vitro fertilization [54]. Notably, all cases of pregnancy loss in individuals with *PADI6* variants corresponded to first-trimester miscarriage rather than willful pregnancy termination [49,55]. This observation is consistent with the early embryonic arrest seen in the aforementioned studies and in mouse models [56,57]. *PADI6* is a part of the subcortical maternal complex (SCMC) that has an important role in genomic imprinting (reviewed in [58]), and *PADI6* mutations cause imprinted disorders [59]. Studies suggest that *PADI6* variants act as either recessive or dominant-negative maternal-effect mutations [60]. Notably, *PADI6* variants cause the same spectrum of pregnancy outcomes as variants in other SCMC component-encoding genes, including chromosomal aberrations and disturbed imprinting [61].

*STIL1* is a centrosomal protein involved in the maintenance of centrosome integrity, mitotic spindle organization, and positioning—essential for centriole duplication during the cell cycle [62]. It was shown that *STIL1* is ubiquitously expressed in proliferating cells and during mouse embryonic development [63,64]. This information is consistent with data in [65], where compound heterozygous variants in *STIL1* resulted in five spontaneous miscarriages occurring between the 7th and 11th gestational week. Similarly to the case of *PADI6*, all of the reported cases of pregnancy loss due to *STIL1* mutations involved spontaneous abortion rather than termination, in good concordance with the role of the gene in early embryogenesis.

The second subgroup includes 9 genes (*DYNC2H1*, *FGFR2*, *FGFR3*, *FRAS1*, *GREB1L*, *LZTR1*, *PIEZO1*, *PIK3R2*, *PTPN11*) associated with severe developmental abnormalities that result in miscarriage or termination based on fetal disorders revealed by ultrasound. For example, the *DYNC2H1* gene encodes a protein involved in ciliary intraflagellar transport, an evolutionarily conserved process that is essential for ciliogenesis and plays a role in cell signaling events. Homozygous and compound heterozygous mutations in the *DYNC2H1* gene have been identified in patients with Jeune asphyxiating thoracic dystrophy and with short rib-polydactyly [66,67]. In the reviewed set of studies, variants in this gene were reported in cases with termination of pregnancy owing to pathologies observed on ultrasound [46] as well as mutations which were prenatally lethal [33].

Another gene in this group, the *FRAS1* gene, encodes an extracellular matrix protein which regulates epidermal-basement membrane adhesion and organogenesis during development [68]. Compound heterozygous missense mutations in this gene may lead to multiple abnormalities including a multicystic dysplastic kidney, distorted ribs and spine, brain defects and bilateral talipes equinovarus in fetus [26]. Knockouts and homozygous mutations in *FRAS1* have been shown to affect kidney and skeletal development in mice and zebrafish [69,70].

*PIK3R2* in humans encodes phosphatidylinositol 3-kinase regulatory subunit beta. A notable feature of this gene is the presence of a recurrent mutation, c.1117G>A (p.Gly373Arg), which was shown to cause a spectrum of related megalencephaly syndromes in a clinical analysis of 42 children [71]. The same recurrent mutation was reported as the cause of a fetal demise in one of the recent studies [52], and the c.1690A>G (p.Lys564Glu) variant has also been implicated in miscarriage [48].ijms-24-17572-t001_Table 1Table 1Pregnancy loss-associated genetic variants in 18 genes reported in several studies.Gene(Locus)Associated DiseasesVariantVariant Origin †Pregnancy OutcomeReference*PADI6*chr1p36.13OMIM:617234NM_207421.4:c.1793A>G (p.Asn598Ser)NM_207421.4:c.2045G>A (p.Arg682Gln)InheritedMiscarriage[55]

NM_207421:c.122C>T (p.Ala41Val)n.a.Miscarriage[49]*STIL*chr1p33OMIM:612703NM_001048166.1:c.1231C>G (p.His411Asp)NM_001048166.1:c.3370A>G (p.Met1124Val)uncertainMiscarriage[65]

NM_001048166:c. 1012C>T (p.His338Tyr)InheritedMiscarriage[51]*DYNC2H1*chr11q22.3
OMIM:613091NM_001080463.1:c.2819-14A4GInheritedTermination[46]
NM_001080463.1:c.7577T4G (p. Ile2526Ser)InheritedTermination[46]

NM_001377.3:c.6047A>G (p.Tyr2016Cys)NM_001377.3:c.6551A>T (p.Asp2184Val)InheritedMiscarriage[33]*FGFR2*chr10q26.13
14 conditionsNM_000141:c.940-1G>An.a.Fetal demise[48]
NM_022970.3:c.764G>A (p.Arg255Gln)Inherited;Neonatal death[34]

NM_022970.3:c.758C>G (p.Pro253Arg)de novoTermination[34]*FGFR3*chr4p16.3
14 conditionsNM_000142:c.1537G>T (p.Asp513Tyr)n.a.Miscarriage[52]
NM_000142:c.742C>T (p.Arg248Cys)n.a.Miscarriage[52]

NM_000142:c.1118A>G (p.Tyr373Cys)n.a.Termination[26]*FRAS1*chr4q21.21
OMIM:219000NM_025074:c.8537C>A (p.Ala2846Asp)InheritedMiscarriage[51]
NM_025074.7:c.1918C>T (p.Arg640Cys)NM_025074.7:c.5205C>A (p.His1735Gln)n.a.Termination[26]*GREB1L*chr18q11.1-q11.2
OMIM:619274NM_001142966.2:c.5614dupA (p.Thr1872Asnfs*)de novoTermination[34]OMIM:617805NM_001142966:c.1305dupA (p.Asp436Argfs*32)n.a.Miscarriage[52]*LZTR1*chr22q11.21
OMIM:616564ENST00000215739.8:c.902G>T (p.Gly301Val)de novoTermination[34]OMIM:605275NM_006767:c.2317G>A (p.Val773Met)n.a.Miscarriage[72]*PIEZO1*chr16q24.3
OMIM:194380NM_001142864:c.1264C>T (p.Gln422Ter)uncertainMiscarriage[72]OMIM:616843NM_001142864:c.2035G>T (p.Glu679X)uncertainTermination[48]

NM_001142864.3:c.3206G>A (p.Trp1069Ter)NM_001142864.3:c.6208A>C (p.Lys2070Gln)InheritedTermination[73]

NM_001142864:c.30_31delAC (p.Leu10fs)uncertainMiscarriage[51]*PIK3R2*chr19p13.11
OMIM:603387NM_005027:c.1117G>A (p.Gly373Arg)n.a.Fetal demise[52]
NM_005027:c.1690A>G (p.Lys564Glu)n.a.Miscarriage[48]*PTPN11*chr12q24.13
4 conditionsNM_002834:c.174C>A (p.Asn58Lys)n.a.Fetal demise[48]
NM_002834.4:c.218C>T (p.Thr73Ile)de novoNeonatal death[34]*COL2A1*chr12q13.11
15 conditionsNM_001844.5:c.3864_3865delCT (p.Cys1289Pfs*)InheritedTermination[34]
NM_001844.5:c.3490G>T (p.Gly1164Cys)de novoMiscarriage[26]*FOXP3*chrXp11.23
OMIM:304790NM_014009.3:c.1009C>T (p.Arg337Ter)InheritedRPL[74]
NM_014009.3:c.906delT (p.Asp303fs*87)InheritedFetal death[75]

NM_014009.3:c.1009C>T (p.Arg337X)InheritedMiscarriage[76]

NM_014009.3:c.1033C>T (p.Leu345Phe)InheritedMiscarriage[77]

NM_014009.3:c.1189CNT (p.Arg397Trp)InheritedMiscarriage[78]

NM_014009.3:c.319_320delTCInheritedMiscarriage[78]*NEB*chr2q23.3
OMIM:619334NM_001164507:c.20974delA (p.Val6993Serfs*8)uncertainMiscarriage[72]OMIM:256030NM_001271208:c.24094C>T (p.Arg8032Ter)NM_001271208:c.20098C>A (p.Leu6700Ile)uncertainMiscarriage[52]*RYR1*chr19q13.24 conditionsNM_000540.2:c.14130-2A>GNM_000540.2:c.9221C>T (p.Ser3074Phe)InheritedTermination[46]

NM_000540.2:c.6721C>T (p.Arg2241Ter)InheritedMiscarriage[79]

NM_000540.2:c.2097_2123del (p.Glu699_Gly707del)InheritedTermination[79]

NM_000540.2:c.7043delGA (p.Glu2347del)InheritedTermination[79]*RYR2*chr1q43OMIM:115000OMIM:115000NM_001035.2:c.409C>T (p.Arg137Trp)NM_001035.2:c.4652A>G (p.Asn1551Ser)InheritedRPL[39]

NM_001035.2:c.12526G>A (p.Val4176Met)InheritedStillbirth[34]*SCN5A*chr3p22.2
9 conditionsNM_001160161:c.3749C>T (p.Thr1250Met)InheritedMiscarriage[51]
NM_198056:c.5393G>A (p.Trp1798Ter)n.a.Fetal demise[52]

NM_198056.3:c.1663G>T (p.Glu555Ter)n.a.Stillbirth[50]

NM_198056.3:c.1858C>T (p.Arg620Cys)n.a.Stillbirth[50]

NM_198056.3:c.5350G>A (p.Glu1784Lys)n.a.Stillbirth[50]*GBE1*chr3p12.2OMIM: 232500OMIM:263570NM_000158:c.467G>A (p.Arg156His)NM_000158:c.-35_-54deluncertainMiscarriage[51]

NM_000158:c.1064G>A (p.Arg355His)InheritedFetal demise[47]

NM_000158:c.1543C>T (p.Arg515Cys)InheritedFetal demise[47]^†^—the following notations are used: uncertain—parental genotypes were not evaluated, but the variant was homozygous or compound in the fetus; n.a.—variant origin unknown.

Variants in *GREB1L*, a gene which plays a major role in early metanephros and genital development, were associated with perinatal lethality and bilateral renal agenesis [80]. In the reviewed set of studies, fetuses with *GREB1L* mutations were aborted or stillborn due to the severity of malformations [34,52,81].

The *PIEZO1* gene encodes a large mechanosensitive ion channel that acts as a nonselective cation channel [82]. *PIEZO1* is involved in a number of crucial processes in the lungs, bladder, colon, and kidney, as well as in sensing of blood flow in the vasculature system [82]. In good concordance with its broad set of functions, *PIEZO1* mutations have been observed in cases with fetal malformations leading to early miscarriage or termination.

Mutations in the leucine zipper-like transcriptional regulator (*LZTR1*) gene were associated with Noonan syndrome (NS)—one of several RASopathies [83]. NS is a genetic disorder characterized by developmental delays, typical facial gestalt and cardiovascular defects. Mutations in multiple genes have been proven to be involved in the disturbance of the transduction signal through the RAS-MAP Kinase pathway and the manifestation of various subtypes of Noonan syndrome. It was shown that *SOS1, RAF1, KRAS, BRAF, NRAS, MAP2K1,* and *RIT1*, and recently described *SOS2*, *LZTR1*, and *A2ML1* are involved in the molecular etiology of this disorder. The first gene described as a causal for NS was *PTPN11* that encodes protein-tyrosine phosphatase, nonreceptor-type 11, which is involved in several developmental processes such as limb development, semilunar valvulogenesis, hemopoietic cell differentiation and mesodermal patterning [84]. Notably, *PTPN11* was also among the pregnancy loss genes reported by multiple studies.

Among genes that serve tissue-specific functions not directly linked to development, one example is *NEB* which encodes giant actin filament-associated protein. Mutations in *NEB* were associated with autosomal recessive nemaline myopathy, a disease characterized by generalized skeletal muscle weakness and the presence of electron dense protein accumulations on patient muscle biopsy examination [85]. Previously, it was shown that severe homozygous truncating *NEB* mutations may result in embryonic lethality [86]. This explains multiple observations of pathogenic *NEB* variants in pregnancy loss.

In the same group of genes, several of them lead to pregnancy loss solely in male fetuses. One of these genes, *FOXP3*, is a transcription factor which controls the activity of genes that are involved in regulating the immune system [87]. Mutations in *FOXP3* cause immune dysregulation, polyendocrinopathy, enteropathy, and X-linked (IPEX) syndrome which is a rare X-linked recessive disorder usually diagnosed in males during infancy and often fatal within the first year of life. At the same time, heterozygous female carriers of *FOXP3* mutations are unaffected and otherwise healthy. Clinical manifestations may be highly heterogeneous in patients with the same mutation, different even within the same family. IPEX syndrome may lead to type 1 diabetes mellitus, hypothyroidism, autoimmune hemolytic anemia, thrombocytopenia, lymphadenopathy, hepatitis, and nephritis [75,88]. Among the reviewed studies, there were cases in which mutations in *FOXP3* led to both miscarriage and intrauterine death or pregnancy termination based on ultrasound results (Table 1). Notably, all of the six cases of pregnancy loss with *FOXP3* mutations corresponded to male fetuses, making *FOXP3* a unique male-specific risk factor of miscarriage. A similar male-specific genetic cause of miscarriage was reported in a Chinese couple suffering from recurrent spontaneous abortion in male fetuses. In this case, WES revealed a novel c.790-6C>T mutation in the *NSDHL* gene. This syndrome is an X-linked dominant condition which leads to the lethality of the male fetuses. Females with the *NSDHL* mutation show phenotypic heterogeneity ranging from a normal to a typical CHILD syndrome phenotype. Studies have indicated that the absence of clinical symptoms may be related to X-chromosome inactivation [89,90].

*SCN5A*, *COL2A1*, *RYR1*, and *RYR2* genes are associated with more than two conditions that affect different body systems, such as cardiovascular, skeletal, muscle and connective tissue. Most likely, the multifunctionality of these genes explains that mutations in them lead to pregnancy loss. It is worth noting that for *COL2A1*, experimental studies showed isoform-specific epigenetic modifications consistent with imprinting [91]. *RYR1* was also reported to be imprinted in some patients with multi-minicore disease which affects muscles used for movement [92]. *RYR2* encodes another ryanodine receptor isoform, a major calcium handling channel located within cardiomyocytes [93]. Mutations in the *RYR2* gene cause the inherited arrhythmia and catecholaminergic polymorphic ventricular tachycardia [93], and, similarly to *SCN5A*, may result in stillbirth.

Mutations in *GBE1*, which encodes the glycogen branching enzyme 1, cause glycogen storage disease type IV. This is a rare metabolic disorder with an autosomal recessive inheritance that involves the liver, neuromuscular, and cardiac systems [94]. It was demonstrated that compound heterozygous mutation in *GBE1* can lead to trophoblastic damage and early fetal loss [95]. Thus, *GBE1* appears to be one of the few cases in which pathogenic variants may have a direct negative impact on the maternal–fetal interface.

It is important to note that the vast majority of the 18 genes described above (with the exception of early embryonic development ones) are associated with inherited diseases that manifest in the postnatal period. This observation indicates that mutations in many of the known PL genes have a heterogeneous phenotypic manifestation, and miscarriage is only one of the possible consequences of these variants in the fetal genotype. On the one hand, this observation is expected given that the identification of causal variants in pregnancy loss is almost inevitably based on known gene–disease relationships. On the other hand, detection of deleterious variants in the same gene both in miscarriage and in patients with monogenic disorders suggests that additional factors determine the actual phenotypic manifestation of these mutations. It may be hypothesized that epistatic genetic interactions or, more likely, the interplay between the maternal and fetal genotypes determines the outcome of pregnancy in these cases.

## 5. Common Properties of Known Pregnancy Loss Genes

The next goal of our review was the analysis of the complete list of 196 known pregnancy loss genes. We tried to systematically characterize the properties of these genes and define the main groups of genes involved in the pathology of pregnancy loss. First, we analyzed the biological processes they regulate via a gene set enrichment analysis method. Gene Ontology (GO) terms as well as Molecular Signatures Database (MSigDB) [96] canonical pathways were used as reference gene sets for this analysis. The analysis performed using the GO biological process terms showed enrichment for genes involved in epithelial tube morphogenesis (27 genes), control of appendage development (17 genes), as well as the urogenital/renal system and heart development (27 and 21 genes) (Figure 2A). In total, only slightly more than a half of PL genes (105) were annotated with development-related GO terms (all GO terms containing the words “development”, “morphogenesis”, and “formation” were considered to be related to developmental processes); at the same time, as many as 91 genes from our list did not correspond to any development-related pathway. This observation confirms our earlier assumption that several distinct mechanisms may lead to pregnancy loss.

In addition to biological process terms, we also investigated the involvement of specific cellular components and proteins with particular molecular functions. Analysis of cellular component enrichment revealed four genes (*ARID1A, ARID1B, SMARCB1, SMARCC2*) among PL genes which are the main components of ATP-dependent chromatin remodeling complexes SWI/SNF (Figure 2B). SWI/SNF (BAF in mammals) are a group of proteins that interact to change DNA packaging and in transcription regulatory complexes. Extensive evidence suggests that SWI/SNF-mediated chromatin remodeling is critical for mammalian embryo development. Down-regulation of *SMARCC2* and *SMARCB1* results in cell cycle arrest and disturbances in gene expression [97]. The expressions of *ARID1A, ARID1B, and SMARCA2* are up-regulated in rhesus monkey blastocyst-stage embryos, implying that these subunits function during cell lineage commitment [98,99]. In good concordance with these findings, enrichment analysis for molecular functions also revealed an overrepresentation of genes involved in transcription (12 out of 188 genes), as well as growth factor (FGF) binding and function (*COL1A1, COL1A2, COL2A1, FGFR2, FGFR3, LIFR, FLT1, NLRP7, SCN5A*) (Figure 2C). The enrichment for FGF-related proteins is expected given the known role of FGF signaling pathways in development; during embryogenesis, FGF signaling plays an important role in the induction/maintenance of the mesoderm and the neuroectoderm, the control of morphogenetic movements, anteroposterior (AP) patterning, somitogenesis and the development of various organs [100,101,102].

Analysis of our gene set against MSigDB curated gene sets (C2) also revealed multiple enriched gene sets (Figure 2D), with one of the the most significant enrichments corresponding to the Ras signaling pathway (14 out of 190 genes). The Ras pathway is one of the best characterized signal transduction pathways in cell biology. The function of this pathway is to transduce signals from the extracellular milieu to the cell nucleus where specific genes are activated for cell growth, division and differentiation. The Ras/Raf/MAPK pathway is also involved in cell cycle regulation, wound healing and tissue repair, and cell migration. Finally, the Ras/Raf/MAPK pathway is able to stimulate angiogenesis through changes in expression of genes directly involved in the formation of new blood vessels [103]. All of the roles played by the Ras/Raf/MAPK pathway in development make its involvement in PL pathogenesis perfectly reasonable. It is worth noting that in Luo et al. studied where screening for novel biomarkers in the endometrium associated with RPL was conducted; differentially expressed genes of RPL patients, compared to control, were also enriched by genes involved in the Ras signaling pathway [104].

Having performed the functional analysis of pregnancy loss genes, we next turned our attention to other properties of the genes that might be used to identify most likely candidate genes for future research. Prior to this analysis, however, we split the whole set of 196 genes into subcategories dependending on the specific features of the cases in which respective causal genetic variants were described (Figure 1). For all the aforementioned gene groups, we assessed several important gene-level metrics that characterize the gene in terms of its functional significance and role in the cell. These included the degree of evolutionary conservation of genes in these groups, as well as the properties of the gene expression pattern (its breadth across tissues and absolute expression levels).

The degree of evolutionary constraint for human genes is commonly assessed using the population frequencies of a specific class of variants that may cause functional knock-outs or knock-downs of a gene (putative loss-of-function (pLoF) variants). Such variants may substantially impact disease risk in their carriers [105]; hence, individuals carrying a pLoF allele likely have lower ability to survive and reproduce in their environment, leaving fewer descendants. Consequently, the frequency of a LoF allele is expected to be lower in the population. Therefore, observing a reduced quantity of pLoF variants in a gene compared to putatively neutral variants is indicative of their deleteriousness [106]. Measures of ’mutation intolerance’ effectively rank genes by the deficit of pLoF variants in large samples [107], and several widely used measures are based on this principle, including pLI [108] and LOEUF [109]. The first metric, pLI, estimates the probability that a pLoF allele present in one copy of the gene causes a haploinsufficient phenotype, and estimates the likelihood that a gene falls into the category of LoF-haploinsufficient genes. pLI separates genes into those intolerant (pLI ≥ 0.9) or tolerant (pLI ≤ 0.1) to LoF [110]. The second metric, ’the loss-of-function observed/expected upper bound fraction’ (LOEUF), represents a conservative estimate of the ratio of observed to expected pLoF variants. Both pLI and LOEUF were initially proposed by the Genome Aggregation Database (gnomAD), and the reference values for human genes are taken from this source.

We performed the evolutionary constraint analysis of the genes involved in PL using the aforementioned pLI and LOEUF metrics. A comparison with all human genes showed that, on average, the genes involved in the pathogenesis of pregnancy loss are more conserved: for these genes, the median LOEUF value was 0.52 (Figure 2F). Curiously, we observed substantial differences for a group of genes found only in cases of non-recurrent miscarriages (non-RPL) compared to other (RPL) genes. For RPL, the proportion of tolerant and intolerant genes was 0.73/0.18, while this ratio was markedly different for non-RPL—0.42/0.46, (Figure 2E). At the same time, the median LOEUF (Figure 2F) value for RPL-associated genes was 0.795, and it was much lower for non-RPL genes (0.376). The value for non-RLP genes was close to the margin of haploinsufficiency, indicating that the genes involved in a single miscarriage tend to be much more evolutionary constrained. A similar trend is seen for genes found in studies where pregnancy resulted in forced termination, with these genes being more conserved than those connected to spontaneous abortion. Comparisons of conservation metrics for SA across trimesters were performed; however, the results are not representative due to the lack of data on gestational age across studies.

To evaluate the gene expression pattern of subsets of genes involved in pregnancy loss, we used the median by-tissue expression data from the Genotype-Tissue Expression (GTEx) consortium (https://www.gtexportal.org/home/, accessed on 1 September 2023) [111], the largest available source of information on gene expression levels in various human tissues. We discovered that pregnancy loss genes are broadly expressed across tissues, with a median value of 29 tissue per gene (Figure 2G). This observation is consistent with a greater degree of evolutionary conservation of these genes. Genes associated with RPL, however, tend to be expressed in fewer tissues (24), though the number is still much higher than for an average gene (which is not expressed in any tissue at the level of 10 TPM). This observation is in perfect consistency with the results of evolutionary constraint analysis, which also showed a lower level of functional significance of these genes. It is also worth noting that causal miscarriage genes are less broadly expressed in tissues (Figure 2G) than pregnancy termination-associated genes, and their median expression level is almost two times lower (Figure 2G,H). For all investigated groups, median expression levels (Figure 2H) across all tissues were from 9 to 16 times higher than for all human genes, highlighting the ubiquitous expression of these genes.

Taken together, our analysis revealed that genes which are involved in the pathogenesis of PL are broadly expressed, highly evolutionarily conserved and involved in crucial cell differentiation and developmental processes and related signaling pathways. At the same time, relatively higher degree of evolutionary conservation and broader expression profile of genes involved in non-recurrent PL may indicate that non-genetic factors play a relatively greater role in RPL. At the same time, comparison between different groups of PL genes may be confounded by biases of individual studies.

## 6. Construction of a List of Candidate Pregnancy Loss Genes

Given the common properties of genes implicated in miscarriage and recurrent pregnancy loss, we next set off to use the acquired knowledge to identify a set of possible candidate genes with a similar set of properties that may be used to search for genetic causes of pregnancy loss in future studies. To achieved this, we used a multi-step strategy which is summarized in Figure 3.

At the first stage, among all human genes, we selected the ones with a degree of evolutionary constraint similar to that of the known pregnancy loss genes (the LOEUF value was below the median value for the studied list of genes). After this, broadly expressed genes were selected among the remaining 5140 ones. For this, we filtered out genes with a median expression across all tissues below the median level of known PL genes (11.2 TPM). This step narrowed the set of candidate genes down to 2777 genes. Next, we analyzed the number of the remaining genes that physically interact with known PL genes. Filtering using known protein–protein interaction data from BioGRID [112] removed firther 475 genes, leaving a broad set of 2302 genes.

To further narrow down this gene list, we went on to select genes involved in pathways we identify during the gene set enrichment analysis (either all pathways or 10 most significant pathways were used in this step). This step resulted in a final set of 207 (for top 10 enriched pathways) or 1886 genes.In the narrow set of 207 genes, 21 overlapped with the known ones, and the remaining 186 could be viewed as confident candidates for investigation in future studies related to pregnancy loss (Appendix A). Out of this set of candidate genes, 103 have previously been reported as causal genes for Mendelian diseases. As many as 83 genes, however, have not yet been reported as causal for any disorder, making them the most important targets for future research.

## 7. Conclusions

A whole range of studies have evaluated the genetic variants which can cause pregnancy loss, including miscarriage, fetal death, stillbirth, or termination of pregnancy for severe fetal malformations. Identification of genetic variants leading to these conditions is extremely important both to deepen the understanding of their pathogenesis and to enable accurate prediction of the risk for each individual couple. Furthermore, accumulation of data supporting the causal role of short genetic variants in pregnancy loss emphasizes the need to change standard practices of genetic testing in pregnancy loss, both spontaneous and especially recurrent. While karyotyping and/or analysis of chromosomal abnormalities are commonly performed, NGS on POCs is still mostly used in research rather than in a clinical setting. Introduction of NGS methods into clinical practice for couples who suffered a loss of pregnancy may greatly enhance family planning, inform the usage of assisted reproductive technologies and consequently decrease the risk of further pregnancy losses for such couples. This is especially important given the deep psychological effects of pregnancy loss on women (demonstrated in multiple studies, e.g. [113]).

Our systematic analysis of NGS-based studies in pregnancy loss revealed a total of 196 genes with point mutations associated with PL, and 18 of these genes were reported as causal in several studies, while others were discovered only once. In-depth evaluation of this gene set suggested that the genes could be classified into several groups depending on the gestational age at pregnancy loss, as well as the molecular functions of the gene product. Further enrichment analysis of the complete list of 196 genes confirmed that these genes are involved in a broad range of pathways. While developmental genes are the dominant group, almost a half of our gene set did not correspond to any developmental biological process. This finding highlights the fact that, while disturbance of essential embryological processes may explain the majority of early miscarriages, there seems to be a plethora of other biological events that may prevent a successful pregnancy outcome. Nevertheless, further research efforts are indeed required both for the analysis of the genetic causes of early pregnancy loss and for creation of an exhaustive list of point mutations leading to all types of pregnancy losses.

In light of the latter goal, in this study, we made an effort to expand the list of candidate genes for further investigation in sequencing-based studies of pregnancy loss, with a total of 83 predicted candidate genes found to have no known disease associations. We believe that such efforts would increase the effectiveness of early detection of risk alleles and prevention of pregnancy loss.

## Figures and Tables

**Figure 1 ijms-24-17572-f001:**
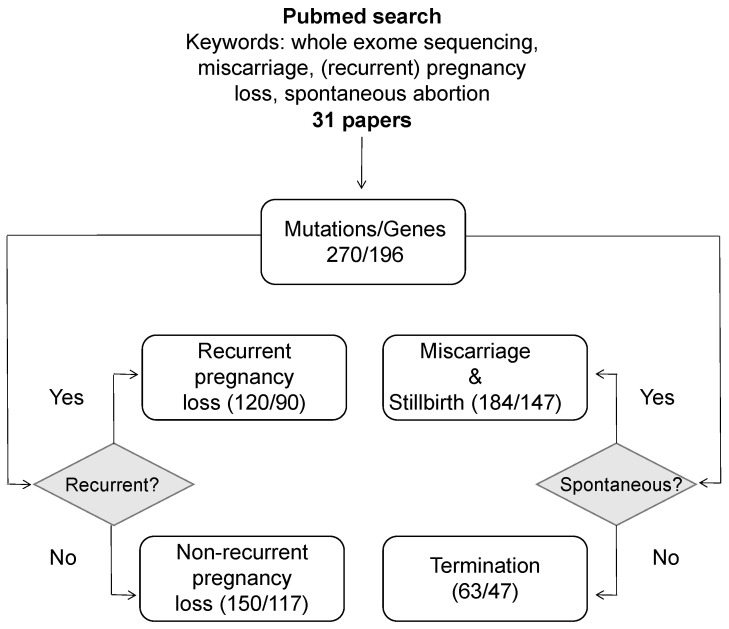
A schematic representation of the workflow used to review known pregnancy loss genes.

**Figure 2 ijms-24-17572-f002:**
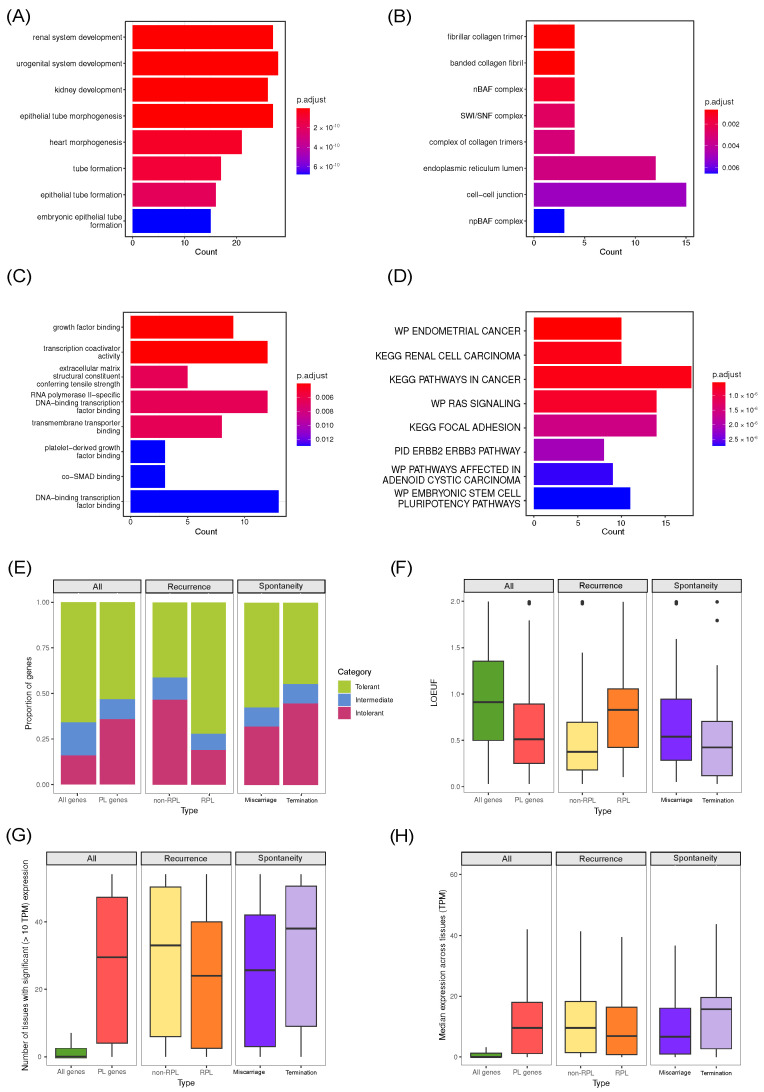
Analysis of the common properties of pregnancy loss genes. (**A**–**D**). Barplots showing gene set enrichment analysis results for the whole set of PL genes using the Gene Ontology terms and MSigDB pathways. (**A**) GO biological processes, (**B**) GO cellular components, (**C**) GO molecular functions, and (**D**) curated gene sets and canonical pathways from MSigDB. The color gradient represents the adjusted significance level. (**E**,**F**) Evolutionary constraint analysis of different subsets of PL genes. In (**E**), genes were grouped into three classes based on pLI value (tolerant, pLI < 0.1; intermediate, 0.1 < pLI < 0.9, and intolerant, pLI > 0.9). In (**F**), boxplots show the gene-level LOEUF values obtained from gnomAD. (**G**,**H**) Boxplots showing the results of gene expression level analysis for different subgroups of PL genes. In (**G**), the number of tissues with expression at the level of at least 10 transcripts per million (TPM) is shown. In (**H**), median expression across tissues is shown. In (**E**–**H**), All genes—all genes in the genome, PL genes—the full set of 196 genes linked to pregnancy loss.

**Figure 3 ijms-24-17572-f003:**
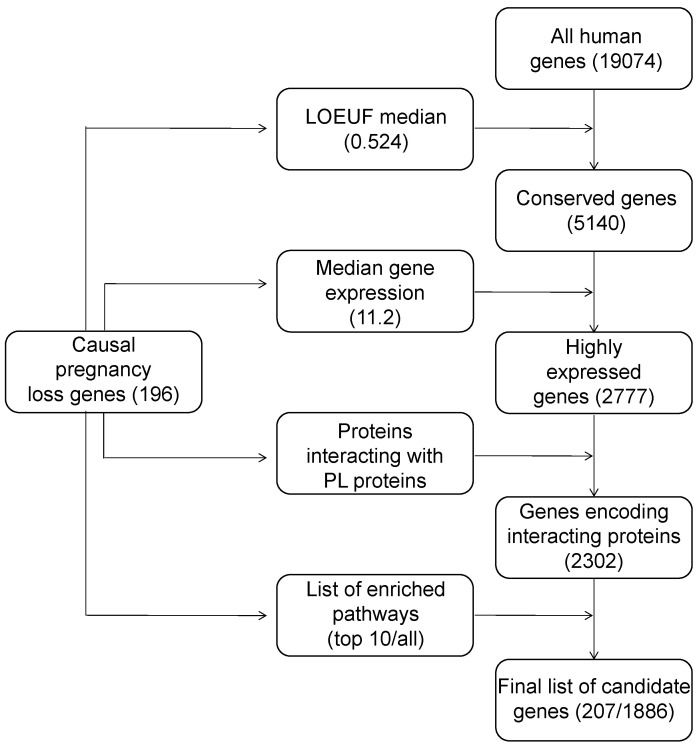
A schematic representation of the workflow used for prediction of candidate genes which may be involved in pregnancy loss. Numbers in barckets represent gene counts.

## Data Availability

Data and code pertinent to the analyses presented in this review are available in the repository at https://github.com/mrbarbitoff/pregnancy_loss_genes.

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
