# Peer review of "The Landscape of Point Mutations in Human Protein Coding Genes Leading to Pregnancy Loss"

_ijms, 2023, doi:10.3390/ijms242417572_

Round 1
Reviewer 1 Report
Comments and Suggestions for Authors
The authors provide a systematic overview of methods used to study the genetic causes of spontaneous abortions and recurrent pregnancy loss in humans. Based on next generation sequencing data, they compile a list of 179 involved genes of which 18 genes have been reported in several studies. Next, the authors provide brief descriptions of the involved genes regarding known functions, gene ontology terms, expression patterns, and evolutionary constraints. Finally, based on a multistep analysis of all human protein coding genes, the authors present a list of 167 candidate genes for pregnancy loss as novel targets for research.
The manuscript is clear, concise, well-organized and well-written and provides valuable overview of the status of knowledge in the field. I have just a few minor comments and edits.
Comments
1. The title could clearly reflect that the review considers protein coding genes only. Maybe “The landscape of point mutations in human protein coding genes leading to pregnancy loss”.
2. Table 1 could include the chromosomal location of these genes.
3. Page 9, lines 359-363: if the described mutation in NSDHL is X-linked dominant, it should affect female fetuses as well. Male fetuses are hemizygous and here dominance/recessiveness does not matter. Please reword.
4. Are any of the described genes imprinted? Please add this information.
5. There is no legend for the Supplementary Table 1 and the two spreadsheets are named in Cyrillic. Please provide legends for both spreadsheets and name the sheets in English.
Minor edits
Page 3, line 125: please remove ‘techniques’ (redundant with ‘technologies’).
Page 4, line 182: OFD1 should be in italics (please check gene symbols throughout the text).
Page 8, line 318: please change Piezo1 to capital letters (unless the authors intentionally reference the murine gene) and italics.
Page 9, line 354: FOXP3 should be in italics.
Author Response
Reviewer #1
The authors provide a systematic overview of methods used to study the genetic causes of spontaneous abortions and recurrent pregnancy loss in humans. Based on next generation sequencing data, they compile a list of 179 involved genes of which 18 genes have been reported in several studies. Next, the authors provide brief descriptions of the involved genes regarding known functions, gene ontology terms, expression patterns, and evolutionary constraints. Finally, based on a multistep analysis of all human protein coding genes, the authors present a list of 167 candidate genes for pregnancy loss as novel targets for research.
The manuscript is clear, concise, well-organized and well-written and provides valuable overview of the status of knowledge in the field. I have just a few minor comments and edits.
Authors: We thank the Reviewer for the positive assessment of our work.
Comments
- The title could clearly reflect that the review considers protein coding genes only. Maybe “The landscape of point mutations in human protein coding genes leading to pregnancy loss”.
Authors: We have changed the title according to suggestions.
- Table 1 could include the chromosomal location of these genes.
Authors: We have added the relevant information about the chromosomal location of genes in Table 1.
- Page 9, lines 359-363: if the described mutation in NSDHL is X-linked dominant, it should affect female fetuses as well. Male fetuses are hemizygous and here dominance/recessiveness does not matter. Please reword.
Authors: We have corrected the information about variants in the NSDHL gene.
- Are any of the described genes imprinted? Please add this information.
Authors: We have added the relevant information about PADI6, COL2A1 and RYR1 genes.
- There is no legend for the Supplementary Table 1 and the two spreadsheets are named in Cyrillic. Please provide legends for both spreadsheets and name the sheets in English.
Authors: Supplementary Table legends were added, and the sheet names were corrected.
Minor edits
Page 3, line 125: please remove ‘techniques’ (redundant with ‘technologies’).
Page 4, line 182: OFD1 should be in italics (please check gene symbols throughout the text).
Page 8, line 318: please change Piezo1 to capital letters (unless the authors intentionally reference the murine gene) and italics.
Page 9, line 354: FOXP3 should be in italics.
Authors: The issues were corrected.
Reviewer 2 Report
Comments and Suggestions for Authors
The authors made a careful review of the literature data relating to fetal loss. The data analysis is accurate and extensive, even if in some points (e.g. relating to chromosomal examination and CMA), it appears redundant and excessively didactic.
Overall, the work is of good quality, and has the main aim of raising awareness towards a more comprehensive search for the genetic causes of fetal losses. I read the work with pleasure and appreciated the effort made by the authors, but the final messages I received are confused.
It requires a radically different approach before publication.
The reason that justifies this comment is the following.
Monofactorial genetic conditions occur on a continuum from the prenatal life, the perinatal period, and early childhood. There is no surprise in the diagnosis of intragenic mutations in association with phenotypic anomalies in the fetus. A fetus with morphologic anomalies is not different from a newborn with morphologic anomalies, even if we must consider that many gene variants observed in the fetus are lethal.
Major criticisms
1) it is required to review the literature data related exclusively to early spontaneous abortion, in which no morphological anomalies of the fetus has been identified, due to limitation of the gestational age or because they were actually absent.
2) it is very useful to provide an overall extimation of the causes of early non-malformative miscarriage, including both genetic (quantitative and qualitative abnormalities), and non-genetic causes
3) after a selective analysis of early abortions, and a comprehensive overview of the underlying causes, a suggestion from the authors about the diagnostic procedure in early pregnancy loos can help.
Author Response
Reviewer #2
The authors made a careful review of the literature data relating to fetal loss. The data analysis is accurate and extensive, even if in some points (e.g. relating to chromosomal examination and CMA), it appears redundant and excessively didactic.
Overall, the work is of good quality, and has the main aim of raising awareness towards a more comprehensive search for the genetic causes of fetal losses. I read the work with pleasure and appreciated the effort made by the authors, but the final messages I received are confused.
Authors: We thank the Reviewer for the positive feedback on our work and usefu
It requires a radically different approach before publication.
The reason that justifies this comment is the following.
Monofactorial genetic conditions occur on a continuum from the prenatal life, the perinatal period, and early childhood. There is no surprise in the diagnosis of intragenic mutations in association with phenotypic anomalies in the fetus. A fetus with morphologic anomalies is not different from a newborn with morphologic anomalies, even if we must consider that many gene variants observed in the fetus are lethal.
Major criticisms
1) it is required to review the literature data related exclusively to early spontaneous abortion, in which no morphological anomalies of the fetus has been identified, due to limitation of the gestational age or because they were actually absent.
2) it is very useful to provide an overall extimation of the causes of early non-malformative miscarriage, including both genetic (quantitative and qualitative abnormalities), and non-genetic causes
3) after a selective analysis of early abortions, and a comprehensive overview of the underlying causes, a suggestion from the authors about the diagnostic procedure in early pregnancy loos can help.
Authors: We thank the Reviewer for these comments. While we agree with the Reviewer on the importance of research into early spontaneous abortions without detectable malformations, we kindly disagree with the necessity of the proposed radical changes to the manuscript structure and content. We would like to provide the motivation for our position below:
- The objective of the present study was to collect and analyze information about genetic variants in the fetus that lead to pregnancy loss. We did not attempt to focus our analysis on a specific group or specific cause of pregnancy loss (such as those with particular gestational age). We would like to argue that the majority of studies dealing with point mutations in pregnancy loss jointly study cases of pregnancy loss due to severe fetal malformations and early spontaneous abortion cases. Earlier efforts to review the subject have also included all studies and cases irrespective of gestational age and the presence of malformations, highlighting the importance of monogenic disease genes in pregnancy loss (e.g., 10.1093/humupd/dmz015). Our work extends these efforts with newer data accumulated over the past few years, and provides an additional overview of the properties of relevant genes and a framework to predict candidate genes:
- The information on the gestational age provided in the studies reviewed in our work was incomplete, and in many cases (where the gestational age was given in the original publication) the values were presented as ranges rather than one specific value for each case. This makes it very hard to clearly identify those cases that correspond to early term spontaneous abortion. If we filter the dataset according to the available information, less than one third (82 out of 270) of genetic variants were identified in early spontaneous abortion cases.
- Regarding the final messaging of our work, we intended to draw the reader’s attention to the necessity of a comprehensive genetic analysis of the causes of pregnancy loss (including both early miscarriage and later miscarriage or termination for fetal pathologies). Specifically, we wanted to highlight the importance and relevance of whole genome or exome sequencing of both parent and fetal genomes, especially in cases of recurrent reproductive losses. Such a genetic testing strategy may help family planning in the affected couples and aiding the use of assisted reproductive technologies..
Your proposal is really interesting and valuable, however, at the moment there is not enough research for such analysis of euploid fetuses with no morphological anomalies and also determines the reasons for the loss, including both genetic (quantitative and qualitative abnormalities), and non-genetic. However, in our further research we will try to address the issue separately.
Despite the aforementioned concerns, we recognize the importance of your comment; hence, we took an effort to improve our manuscript in several ways, as follows:
- We have expanded the introduction section of the review to include more details on pregnancy loss causes and the role of fetal malformations. We also briefly discuss the role of maternal and fetal genotypes in the pathogenesis of the condition (pages 1-2);
- We now include the discussion of gestational ages at which the pregnancy loss occurred in the reviewed studies (p. 6, lines 250-252);
- We have rewritten the Conclusions section of the work to improve the final messaging, and emphasized the clinical implications and the importance of research into the genetics of early spontaneous abortions with no fetal malformations (pages 14-15)